

# Brief communication: Post-event analysis of loss of life due to hurricane Harvey

Sebastiaan N. Jonkman[1], Maartje Godfroy[1], Antonia Sebastian[1,2], Bas Kolen[3,4]

[1]Department of Hydraulic Engineering, Faculty of Civil Engineering & Geosciences, Delft University of Technology, Delft, the Netherlands
[2]Department of Civil & Environmental Engineering, Rice University, Houston, Texas 77005
[3]Department of Values, Technology, and Innovation, Faculty of Technology, Policy, and Management, Delft University of Technology, Delft, the Netherlands
[4]HKV Consultants, Lelystad, the Netherlands

*Correspondence to*: S.N. Jonkman (S.N.Jonkman@tudelft.nl)

**Abstract.** An analysis was made of the loss of life directly caused by hurricane Harvey. Information was collected for 70 fatalities that occurred directly due to the event. Most of the fatalities occurred in the greater Houston area, which was most severely affected by extreme rainfall and heavy flooding. The majority of fatalities in this area were recovered outside the designated 100 and 500 year flood zones. Most fatalities occurred due to drowning (81%), particularly in and around vehicles. Males and people over 50 years old were overrepresented.

## 1. Introduction and background

Loss of life is one of the most critical consequences of floods. Previous analyses of flood fatalities have focussed on certain regions (Coates, 1999; Zahran et al, 2008; Ashley and Ashley, 2008) and the causes and circumstances of fatalities based on aggregated data for multiple smaller events (Jonkman and Kelman, 2005). Documentation of flood events with significant life loss is scarce and limited to a few cases, such as the flooding of New Orleans due to Hurricane Katrina (2005) (Jonkman et al., 2009) or more general data for large-scale events such as cyclones in Bangladesh (1991) (Chowdhury et al., 1993). However, detailed analysis of mortality during past flood events is important for the development and improvement of consequence and risk models, and to inform policies and public communication.

On August 25, 2017, Hurricane Harvey made landfall near Rockport, Texas as a Category 4 storm with maximum sustained winds of approximately 200 km/hour. Harvey caused severe damages in coastal Texas due to its extreme winds and storm surge, but will go down in history for record-setting rainfall and flood-related damages in southeast Texas. Rainfall totals during the six-day period between August 25 and 31, 2017 were amongst the highest ever recorded in U.S. history. Across large portions of Harris County, rainfall exceeded 1000 mm. The highest recorded 3-day rainfall total was 1318 mm (51.18 inches) in Cedar Bayou near Baytown, Texas (HCFCD, 2017). In this area, the rainfall total during Harvey has been estimated to have exceeded a 9,000-year return period, far above the standard design criteria (Oldenborgh et al., 2017).



The City of Houston and surrounding areas experienced unprecedented urban flooding. Many of Houston's creeks and bayous exceeded their channel capacities, reaching water levels never before recorded, and widespread pluvial flooding was reported in communities across the greater Houston area on the morning of August 27. While the floodwaters in most areas receded within 24-48 hours after the heaviest rainfall, in a smaller number of areas (e.g. downstream of Addicks and Barker reservoirs),

water levels remained elevated for a period of days to weeks. It is estimated that more than 80,000 homes were affected by flooding of at least 46 cm (18 inches) (FEMA, 2017).

During the period between August 25 and 31, rising waters necessitated major rescue operations. In total, more than 120,000 people were rescued by professional and volunteer rescuers (FEMA, 2017). No large-scale mandatory evacuation was ordered before or during Harvey, as the risks of evacuating millions of people were considered too high. Instead, people were advised

to shelter at home and to prepare themselves. However several local evacuations were ordered during the event for areas with specific risks and circumstances, e.g. downstream of Addicks and Barker Reservoirs in the Buffalo Bayou watershed.

As part of a larger rapid fact-finding effort in the wake of Hurricane Harvey (Sebastian et al., 2017), a dataset of flood fatalities was compiled. The objective of this paper is to analyse the causes and circumstances of the fatalities due to Hurricane Harvey.

## 2. Data and methods

In order to analyse the Harvey-related fatalities, a database of reported flood fatalities was compiled (Godfroy et al., 2017). Information about the victim (age, gender) and the circumstances of death (location, time of recovery, cause and circumstances of death) were included.

The analysis focussed on direct fatalities due to Harvey. The database is limited to victims that were recovered within the first two weeks after landfall (August 25-September 8, 2017). Some of the fatalities due to drowning were recovered in the second

week after landfall after the floodwaters receded.

The dataset was compiled using both official government sources and media sources. In the days after Hurricane Harvey, information about casualties appeared mostly on local news websites. Official reporting on casualties and the level of detail varied greatly between counties. Harris County, where most of the casualties occurred, is the only county so far that has published an official and public list of casualties (IFS, 2017).

## 3. Results

Based on our analysis, at least 70 deaths can be directly attributed to hurricane Harvey. The fatalities occurred across 14 counties with a large spatial scatter throughout southeast Texas. Approximately half of the casualties were located in Harris County (37 out of 70). Six fatalities occurred in Orange county around the city of Beaumont and in Galveston county south of Houston. The recovery locations of the victims are shown for the state of Texas (fig. 1) and for Harris County which covers a

large part of the Houston area (fig. 2). For Harris County the 100 and 500-year floodplains have been added using the FEMA





National Flood Hazard layer available through the Texas Natural Resources Information System (TNRIS) (TNRIS, 2017). Of the 37 casualties in Harris County, 18 were located inside the designated 500 year floodplain, of which 8 in the 100 year flood plain. Four casualties occurred in hospitals which are not located in the designated floodplains. The majority of all fatalities (80%) occurred within the first seven days after landfall.

Figure 3 shows an overview the causes and circumstances of death. Fatalities appeared to be mostly caused by drowning (57 out of 70; 81%). Many of the casualties were discovered after floodwaters had receded. A significant proportion of the victims drowned as a result of driving a vehicle into floodwaters or getting swept away by the current while getting out of a car (confirmed for 21 of 57 drowning victims). A tragic example is a case were six individuals of one family, including four children, drowned when their van was swept off a flooded road by the current and ended up in high water in east Houston. A

small number of victims (8 of 57 drowning casualties) were found inside buildings after the flood water receded. Six people drowned when their boat capsized, while they were trying to rescue residents in need from their flooded houses. 28% of the drowning victims (16 out of 57) were recovered in the open and no further information on the exact circumstances was available.

Besides drowning, electrocution and lack of medical treatment were the second largest cause of fatalities during Harvey (both

four victims). This latter category covers fatalities that occurred when (very) ill people were not able to gain access to proper treatment in time (i.e. dialysis or treatment of asthma and heart conditions). A smaller number of fatalities occurred due to other causes such as physical traumas (i.e. car accidents, falling trees) (n=3) or infection due to contact with contaminated flood water (n=1)

Of the victims, 70% were male and 30% were female. This distribution is in line with findings for previous floods. An

important cause could be that men tend to be involved more in risky activities, such as driving and rescue (Jonkman and Kelman, 2005).

The age distribution has been considered. 9% of the fatalities were younger than 18, whereas 56% of fatalities were older than 50 years and 29% older than 65. This suggests an increased vulnerability for elderly people as was also found for the flooding of New Orleans due to Hurricane Katrina (Jonkman et al., 2009).


## 4. Discussion

The previous analyses are based on data gathered from a combination of official public sources and media sources. When only information from the official source for Harris County was utilized (N=37), findings with respect to the importance of drowning (92%), and the over-representation of males (73%) were still obtained.

The present analysis is based on a dataset that includes 70 direct fatalities due to Harvey. Smith et al. (2017) reported 84 fatalities for the event. A difference could be due to the selection criteria. Local media have reported at least ten other casualties. These were not included in the present database when the relationship with Hurricane Harvey could not be confirmed by authorities or when no additional details on the victim or incident were released.



Of the 37 casualties in Harris County, only 22% was in the 100 year flood plain – used as the primary delineation of flood risk in the United States – and 49% within the 500 year floodplain. It is noted that the flooding during Hurricane Harvey is expected to have a much higher return period in the greater Houston region, thus flooding can be expected beyond the 100-year and even 500-year flood zone. However, previous research has also shown that in the greater Houston region, the 100-year

floodplains are a poor predictor of the location of damaging flooding during past events (Highfield, Norman, and Brody, 2013; Blessing, Sebastian, and Brody, 2017) and that NFIP claims can occur far outside of the delineated flood hazard zones (Brody et al. 2015). As future work it is recommended to recreate and map the flood conditions during Harvey based on modelling and field observations to better understand the flood conditions at the time and location of fatalities.

Previous studies have concluded that the majority of fatalities during Atlantic tropical cyclone events occur due to flooding,
particularly due to storm surge (Rappaport, 2014). During Harvey, most fatalities were associated inland flooding in the greater Houston area, primarily driven by Harvey's extreme rainfall. Storm surge of about 2 to 3 meters mainly affected local coastal areas in south Texas and two fatalities were reported in this area.

Findings for Harvey can be compared to other events that affected the same region. During Hurricane Ike (2008) only 11% of the 74 fatalities were due to drowning (Zane et al., 2011). Others were due to injuries and illnesses, partly also related to the
mass-evacuation. In terms of rainfall-driven flooding there are more similarities to Tropical Storm Allison (2001). Tropical Storm Allison dropped more than 762 mm (30 inches) of rainfall over the eastern portion of Harris County June 5-9, 2001. The event led to 41 fatalities, of which 23 occurred in Texas (Stewart, 2001). Twenty-seven deaths were attributed to drowning from freshwater flooding during T.S. Allison. At the time it was the costliest and deadliest urban flood in U.S. history.

It is estimated that more than 100,000 people were directly affected by floodwaters from Harvey. The overall mortality is thus
smaller than 0.1% which is an order lower than typical mortality values reported for storm surge flooding which are around 1%, such as for New Orleans (Jonkman et al., 2009). However, the flooding of Houston is expected to be less deadly than those in New Orleans during Katrina for a number of reasons: flood depths and flow velocities in most areas were lower, waters receded more quickly, and people were warned for flooding beforehand. Still, during Harvey the great majority of fatalities occurred due to drowning, especially in and around vehicles, but relatively few fatalities (11% of the total) were
found in buildings.

The findings also provide a basis for policy recommendations. Following previous studies (Kolen, 2013; Jonkman and Kelman, 2005; Drobot et al., 2007), it is clear that driving on flooded roads is extremely dangerous and several risks are associated with rescue clean up and recovery (e.g. electrocution, CO poisoning). Better identification and communication of 'high-risk' areas for drowning and low-water crossings is required. During future events, preventive closure of floodable roads and underpasses
could be considered. Also, preventive evacuation of could be considered of selected areas where particularly dangerous flood conditions are expected.

The present study focussed on direct fatalities during the event. Analysis and recording of longer term health impacts is recommended. Finally, standardized data collection for future events is recommended to provide a wider data basis and better information for prevention and public communication to avoid future flood fatalities.





## 5. Data availability

The database that was used is publicly accessible at the data repository of the Dutch 4TU.federation (Godfroy et al., 2017).

## 6. Acknowledgements

S.N. Jonkman acknowledges support from the Dutch NWO TTW projects SAFElevee (project number 13861) and All Risk
(P15-21). The contributions by Paul Risher and Jason T. Needham (USACE HEC) are gratefully acknowledged.

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





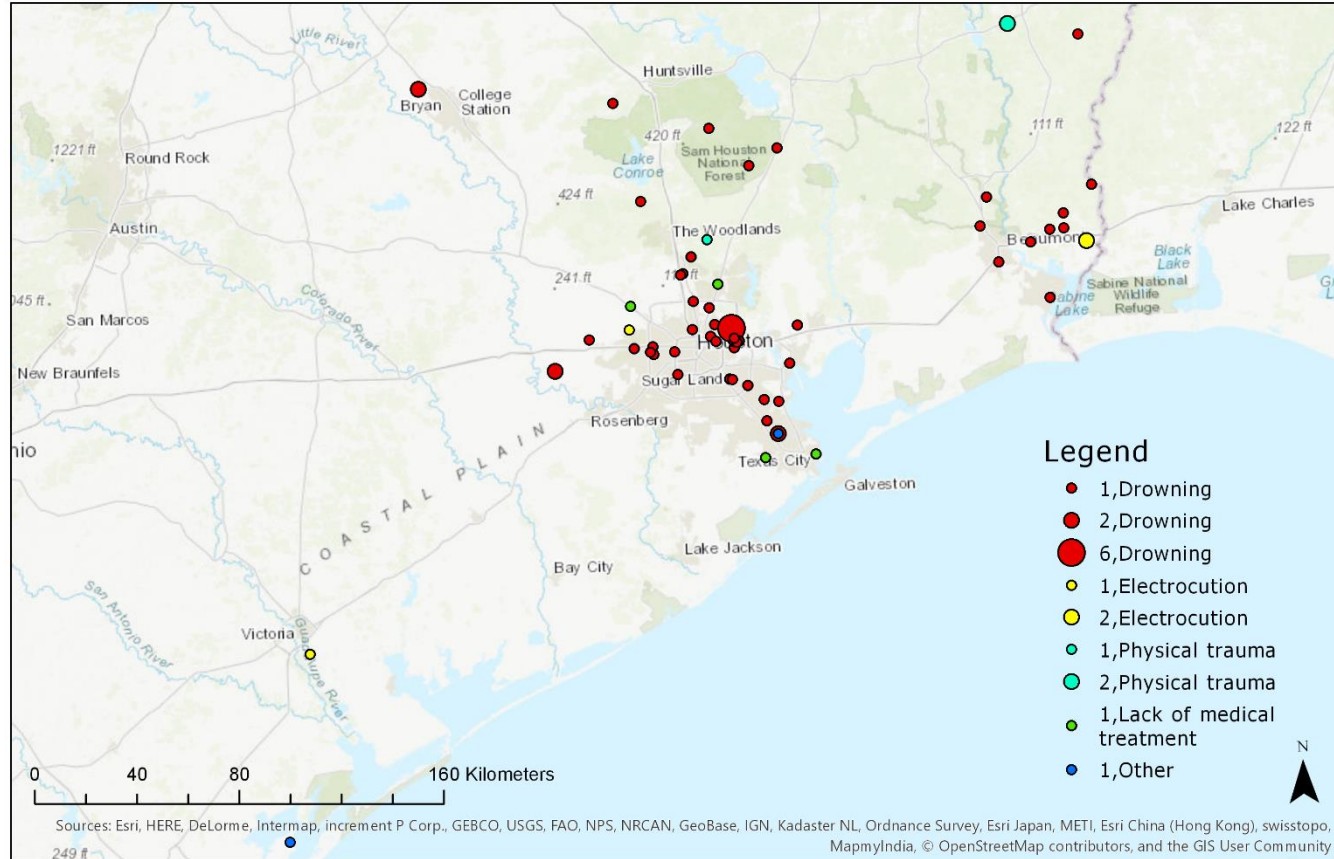

**Figure 1: Location and cause of fatalities due to hurricane Harvey in Texas as of September 8, 2017.**





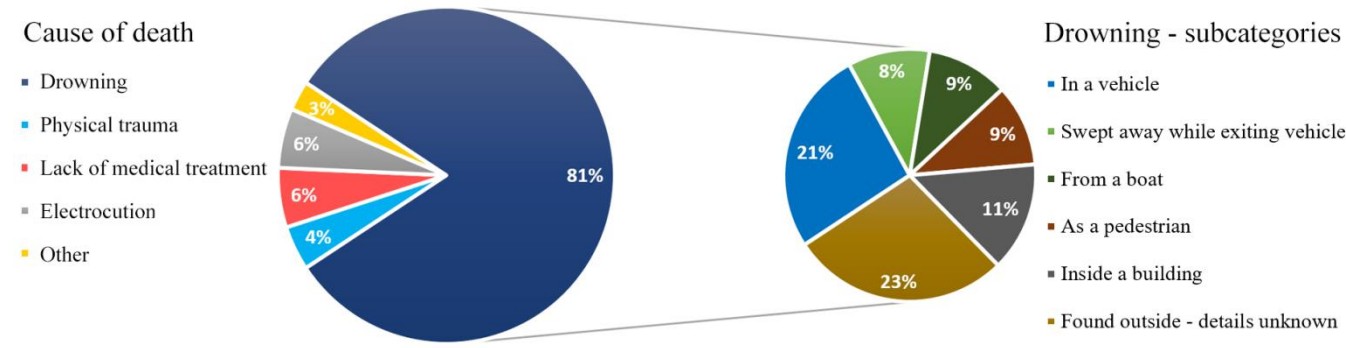

**Figure 2. Location and cause of fatalities due to hurricane Harvey in the Houston metropolitan area (Harris County) as of September 8, 2017. For the victims who died in the hospital (4), the location of the hospital is shown.**

**Figure** 3**. Causes of death for hurricane Harvey (n=70).**

