# Peer review of "Brief communication: Loss of life due to Hurricane Harvey"

_Natural Hazards and Earth System Sciences, 2017_

## Referee Comment (RC1) · I. Kelman (Referee) · 21 Jan 2018

This is an excellent Brief Communication, well-worthy of publication in NHESS given its topical importance, robust methodology, sound analysis, and helpful conclusions. The authors describe exactly what they did, why they did it, what they could not do, and relationship to other sources and analyses including logical, defensible explanations. I request only some minor clean-up points and clarifications prior to publication (and the paper requires a mild copy edit): 1. In the "other" category, Figure 1 seems to have one sample and Figure 3 seems to have two samples (3%). Surely the exact cause of death could be listed rather than "other"? 2. "the risks of evacuating millions of people were considered too high". By whom? There was an intense debate at the time regarding evacuation. Furthermore, many immigrants, legal or otherwise, indicated that they

would have considered evacuating, but they were scared by the lack of guarantee from the government that they would be permitted through checkpoints when evacuating. Please add one sentence here describing exactly who felt that evacuation risks were too high, indicating that some people wanted to evacuate but had reasons not to. 3. Rather than the word "victim", it would be better to repeat "fatality", "death", or similar. 4. The distinction between direct and indirect deaths is difficult to make. In this paper, I can see exactly why the terms "direct" and "directly" are used and I could understand if the authors might be reluctant to remove these terms. Instead, perhaps either use "immediate deaths" or else add one sentence clarifying what is meant by "direct" and "directly" while indicating the reticence of many disaster deaths researchers to distinguish between direct and indirect deaths. 5. "Approximately half of the casualties were located in Harris County". It is implied, but not stated explicitly, that the dominance of Harris County is due to better records. Please make a short, explicit statement. 6. Is there any material covering whether or not the boat fatalities were wearing PFDs (lifejackets)? Please note this information or just indicate that it is not known. 7. When return periods are given, please indicate the parameter being calculated (e.g. flood depth, volumetric flow rate, areal extent). 8. Delete "At the time it [Allison] was the costliest and deadliest urban flood in U.S. history." I think that Johnstown 1889 counts as an urban flood in U.S. history. Allison might not even count as the deadliest urban flood in Texan history depending on how "urban" is defined. 9. "people were warned for flooding beforehand". They were warned before Katrina also. Then, "during Harvey the great majority of fatalities occurred due to drowning, especially in and around vehicles": Could there have been any chance these fatalities occurred during evacuations, i.e. people trying to leave after receiving a warning? The point here is to be wary of assumptions that warnings inevitably reduce death toll, because the social process of warning systems is not straightforward. 10. For the title, perhaps simply "Loss of life from Hurricane Harvey". Finally, congratulations to the authors for not mentioning climate change, demonstrating how they understand the fundaments of vulnerability to Hurricane Harvey. Please ensure that climate change remains absent from this paper,

because it was vulnerabilities which caused the deaths, not climate change. None of the above points detract from the relevance, originality, and worthwhileness of this article which I hope to see published soon. The authors have done a useful and needed analysis on an important case study, providing a solid baseline for further work.

---

## Referee Comment (RC2) · Anonymous Referee #2 · 22 Jan 2018

General comments: This short manuscript provides an assessment of flood-related deaths due to Hurricane Harvey. The mortality data are gathered from government and media sources, revealing the general location, or circumstance, of death, as well as age and gender of the victims. Additional spatial analysis determines the proportion of fatalities that occurred within FEMA-derived flood zones; interestingly, many of the victims were recovered outside of standard delineations of the 100 and 500-year flood zones.

Broadly, this is a well written manuscript on a significant impact marker from Harvey. I have no major issues with the structure of the manuscript, or methods used. My only minor issue is related to the lack of injection of prior literature/findings on specific vulnerabilities such as age and gender (see

http://www.ilankelman.org/disasterdeaths.html for a robust list); also, more discussion on flood/hurricane evacuation literature could be provided. Alas, this is a short communication so I'm not sure how much more should be expected.

Specific comments: Page 3, line 1; Page 4, lines 1-5: A word of caution should be presented here regarding the use of FEMA-designated zones. When were these data valid for? Though the data were acquired in 2017, how long ago where the zones mapped and approved by FEMA? Are these the regulatory zones in NFIP or the same data that is presented in HAZUS? A reference to recent research on the differing delineations of these zones could be provided to make sure that the reader appreciates that the Federal zones are often thought of as "conservative"; see: https://link.springer.com/article/10.1007/s11069-017-2806-6

Page 3, line 15: The parenthetical about victims is confusing.

Page 3, line 23: Rewrite and incorporate this very short sentence here with the next line.

Figures: Percent(s) between Fig. 1 and 3 are off a bit.

In defining direct and indirect deaths, you may want to examine the NOAA Storm Data directive: http://www.nws.noaa.gov/directives/sym/pd01016005curr.pdf

---

## Referee Comment (RC3) · Anonymous Referee #3 · 12 Feb 2018

This brief communication analyses flood-related fatalities due to Hurricane Harvey, based on a database developed by two of the authors of the article. Data were collected from a number of different media and official sources, and include information such as location, circumstances and cause of death. The manuscript briefly introduces the event, the database construction methodology, and finally describes and discusses the main findings.

This is a very relevant and potentially useful piece of work. The database itself is thorough and impeccably organized, and the manuscript is well structured and written. Given that this is a brief communication, intended to be short and concise, I think the authors have done a great job, and this is certainly a worthy contribution for NHESS. In my opinion, no major changes are required.

[Figure]

The only minor issue I have is with some of the statements made in the Discussion section, P4 L26-31. Here the authors state that some of the findings may be used as a basis for policy recommendations, which in principle is fine. However, the following statements feel too vague, not supported by facts, and therefore unscientific. Specifically: "Better identification and communication of 'high-risk' areas for drowning and low-water crossings is required. During future events, preventive closure of floodable roads and underpasses could be considered. Also, preventive evacuation of could be considered of selected areas where particularly dangerous flood conditions are expected." All of this is more or less common sense; nothing is ever perfect and can always be improved. However, are these recommendations relative to this specific case study? If so, the reader is left in the dark regarding whether some of these measures were or not in place in the affected municipalities (or if they were or not "considered"), as well as the whys and why nots. Just looking at the causes of death without taking into account the actual local context seems unsuitable as a basis for policy recommendations. Regarding evacuation, it would also seem that the authors slightly contradict themselves, as in P2 L10-11 the following is stated: "No large-scale mandatory evacuation was ordered before or during Harvey, as the risks of evacuating millions of people were considered too high. Instead, people were advised to shelter at home and to prepare themselves. However several local evacuations were ordered during the event for areas with specific risks and circumstances, e.g. downstream of Addicks and Barker Reservoirs in the Buffalo Bayou watershed." From these statements, it appears that local authorities did in fact consider the evacuation of selected areas. There might have been a reason for other areas not having been evacuated that we do not know about. What I feel is that this paragraph opens a can of worms, and the policy recommendations issue cannot be fully addressed within this type of short manuscript. On the other hand, I recognize that this is something that might be interesting to mention. In my opinion, if this paragraph is to be included, it should be reworked to address the above concerns.

Specific comments:

P2 L16: "victim" should read "victims"

P2 L16: Here I'd suggest replacing "circumstances of death (location, time of recovery, cause and circumstances of death)" with simply "location, time of recovery, cause and circumstances of death", to avoid repeating "circumstances of death" outside and inside the parentheses, which sounds incorrect.

---

## Author Comment (AC1) · 9 Mar 2018

Response to reviewer comments

We acknowledge the valuable comments of the reviewers. We respond to all comments below and have made several changes based on these comments. A few more generic issues are discussed below. Both reviewers (Ilan Kelman and anonymous reviewer nr. 3) raise the issue of defining direct and indirect deaths. We agree that this is difficult and sometimes confusing issue and have adopted the classification proposed by the National Weather Service (2016) and have changed the formulation in various parts of the paper accordingly. Most fatalities in our analysis are due to direct causes, in this case drowning (81%). We also note that the study is limited to

information on fatalities recovered in the first two weeks after Harvey and we recommend to collect information on the longer fatalities and health impacts of the event. Both reviewers Ilan Kelman and anonymous reviewer 3 refer to issues associated with evacuation. We have not found specific evidence of issues related to evacuation of undocumented immigrants or other vulnerable groups, so have not included this in the paper. However, the following press article described some of the challenges in this area: https://www.washingtonpost.com/local/immigration/for-houstons-many-undocumented-immigrants-storm-is-just-the-latest-challenge/2017/08/28/210f5466-8c1d-11e7-84c0-02cc069f2c37_story.html?utm_term=.f8d111c52eb4 Reviewer number 3 comments on the evacuation (orders) by local authorities. Indeed, during Harvey there seemed to be no mass evacuation order, but during the course of the event several local evacuations were ordered for areas with specific risks and circumstances, e.g. in Fort Bend County near the Brazos River. Within the scope of this (short) paper it is not feasible to describe and evaluate the evacuation practices and their effect on life loss during Harvey. However, we have changed the discussion on the last part of the paper and have a) mentioned the dependence between evacuation strategy and life loss; b) recommended to evaluate evacuation and emergency management performance based on the experiences during Harvey as future work.

  Commments by Ilan Kelman

1. In the "other" category, Figure 1 seems to have one sample and Figure 3 seems to have two samples (3%). Surely the exact cause of death could be listed rather than "other"?

Response: we have modified figure 1, so that it is visible that the fatality in the other category occurred in the lower left side of the map

2. "the risks of evacuating millions of people were considered too high". By whom? There was an intense debate at the time regarding evacuation. Furthermore, many immigrants, legal or otherwise, indicated that they would have considered evacuating,

but they were scared by the lack of guarantee from the government that they would be permitted through checkpoints when evacuating. Please add one sentence here describing exactly who felt that evacuation risks were too high, indicating that some people wanted to evacuate but had reasons not to.

Response: we have clarified that the decision about the evacuation was made by the mayor of Houston. See also: https://edition.cnn.com/2017/08/27/us/houston-evacuation-hurricane-harvey/index.html

We could not obtain more specific (and official) information on the evacuation issues for immigrants, although this press source seems to highlight these issues as well: https://www.washingtonpost.com/local/immigration/for-houstons-many-undocumented-immigrants-storm-is-just-the-latest-challenge/2017/08/28/210f5466-8c1d-11e7-84c0-02cc069f2c37_story.html?utm_term=.f8d111c52eb4

3. Rather than the word "victim", it would be better to repeat "fatality", "death", or similar.

Response: we have now used fatality throughout the paper.

4. The distinction between direct and indirect deaths . . . . . . . . . . . . . . Response: see first page

5. "Approximately half of the casualties were located in Harris County". It is implied, but not stated explicitly, that the dominance of Harris County is due to better records. Please make a short, explicit statement.

Response: wording altered to reflect this. This is the most densely populated county in the affected areas and it was severely affected by flooding

6. Is there any material covering whether or not the boat fatalities were wearing PFDs (lifejackets)? Please note this information or just indicate that it is not known.

Response: done

7. When return periods are given, please indicate the parameter being calculated (e.g. flood depth, volumetric flow rate, areal extent).

Response: we refer to the areal extent, short explanation added.

8. Delete "At the time it [Allison] was the costliest and deadliest urban flood in U.S. history." I think that Johnstown 1889 counts as an urban flood in U.S. history. Allison might not even count as the deadliest urban flood in Texan history depending on how "urban" is defined.

Response: sentence removed

9. "people were warned for flooding beforehand". They were warned before Katrina also. Then, "during Harvey the great majority of fatalities occurred due to drowning, especially in and around vehicles": Could there have been any chance these fatalities occurred during evacuations, i.e. people trying to leave after receiving a warning? The point here is to be wary of assumptions that warnings inevitably reduce death toll, because the social process of warning systems is not straightforward.

Response: we have no evidence that these fatalities occurred during evacuation, Some anecdotal evidence and press articles show that fatalities were rescuers or public officials who ended up in the floodwater. We have mentioned the effect of evacuation and individuals behaviour on life loss in a broader context in the last paragraph.

10. For the title, perhaps simply "Loss of life from Hurricane Harvey". Response: title changed and shortened

Reviewer number 2

Reviewer number 2 requests more literature on specific vulnerabilities such as age and gender and flood and hurricane evacuation literature. Response: since this is a short communication we cannot add many more references. We have added two references related to flood evacuation and the link with life loss (Parker et al and French et al). Vulnerability factors (age, gender) are already addressed in several

of the references included (e.g Jonkman and Kelman 2005 and Ashley and Ashley 2005). Specific comments: Page 3, line 1; Page 4, lines 1-5: A word of caution should be presented here regarding the use of FEMA-designated zones. When were these data valid for? Though the data were acquired in 2017, how long ago where the zones mapped and approved by FEMA? Are these the regulatory zones in NFIP or the same data that is presented in HAZUS? A reference to recent research on the differing delineations of these zones could be provided to make sure that the reader appreciates that the Federal zones are often thought of as "conservative"; see: https://link.springer.com/article/10.1007/s11069-017-2806-6 The floodplain maps used for Harris County are the 2016 Effective Floodplains; however, it is important to note that the majority were mapped as part of the Tropical Storm Allison Recovery Project (TSARP) completed in 2007. Frequent updates have occurred for portions of these maps, but we acknowledge that they may underestimate flooding due to changes in land use/land cover, subsidence, and rainfall intensity and duration. Page 3, line 15: The parenthetical about victims is confusing. Response: wording changed Page 3, line 23: Rewrite and incorporate this very short sentence here with the next line. Response: wording changed Figures: Percent(s) between Fig. 1 and 3 are off a bit. Response: see response to the first comment by Ilan Kelman

Reviewer number 3

This reviewer argues that our recommendations were too broad / vague. We have made the recommendations more specific and have linked them to the findings of the paper. We have also added recommendations for a thorough evaluation of evacuation and emergency management during Harvey. Detailed comments: P2 L16: "victim" should read "victims" – Response: changed P2 L16: "victim" should read "victims" P2 L16: Here I'd suggest replacing "circumstances of death (location, time of recovery, cause and circumstances of death)" with simply "location, time of recovery, cause and circumstances of death", to avoid repeating "circumstances of death" outside and inside the parentheses, which sounds incorrect. Response: agreed, wording changed.